# Adolescent health and well-being check-up programme in three African cities (Y-Check): protocol for a multimethod, prospective, hybrid implementation-effectiveness study

Prerna Banati [ID],[1] David Anthony Ross,[2] Benedict Weobong,[3] Saidi Kapiga,[4,5] Helen Anne Weiss [ID],[5] Valentina Baltag [ID],[1] Farirai Nzvere [ID],[5,6] Franklin Glozah,[3] Mussa Kelvin Nsanya [ID],[4] Giulia Greco,[5] Rashida Ferrand,[6,7] Aoife Margaret Doyle [ID],[5,6] The Y-Check Research Programme Team

For numbered affiliations see end of article.

**Correspondence to**
Prerna Banati; banatip@who.int

## ABSTRACT

**Background** During adolescence, behaviours are initiated that will have substantial impacts on the individual's short-term and long-term health and well-being. However, adolescents rarely have regular contact with health services, and available services are not always appropriate for their needs. We co-developed with adolescents a health and well-being check-up programme (Y-Check). This paper describes the methods to evaluate the feasibility, acceptability, short-term effects and cost-effectiveness of Y-Check in three African cities.

**Method** This is a multi-country prospective intervention study, with a mixed-method process evaluation. The intervention involves screening, on-the-spot care and referral of adolescents through health and well-being check-up visits. In each city, 2000 adolescents will be recruited in schools or community venues. Adolescents will be followed-up at 4 months. The study will assess the effects of Y-Check on knowledge and behaviours, as well as clinical outcomes and costs. Process and economic evaluations will investigate acceptability, feasibility, uptake, fidelity and cost effectiveness.

**Ethics and dissemination** Approval has been received from the WHO (WHO/ERC Protocol ID Number ERC.0003778); Ghana Health Service (Protocol ID Number GHS-ERC: 027/07/22), the United Republic of Tanzania National Institute for Medical Research (Clearance No. NIMR/HQ/R.8a/Vol.IX/4199), the Medical Research Council of Zimbabwe (Approval Number MRCZ/A/2766) and the LSHTM (Approval Numbers 26 395 and 28312). Consent and disclosure are addressed in the paper. Results will be published in three country-specific peer-reviewed journal publications, and one multicountry publication; and disseminated through videos, briefs and webinars. Data will be placed into an open access repository. Data will be deidentified and anonymised.

**Trial registration number** NCT06090006.

## BACKGROUND

To unlock human potential and accelerate progress towards achieving the Sustainable Development Goals, it is essential to improve the health and well-being of adolescents (10–19 years).[1] Health is an essential component of human capital,[2] yet adolescent investments have focused primarily on either health or education services with little attention to synergies between these.[3] Research investments in the first 1000 days of life have dramatically outweighed investments in the subsequent 7000 days, leaving an evidence gap on how to develop and sustain human potential through adolescence and early adulthood.[4]

Among adolescents in low-income and middle-income countries (LMICs), HIV/AIDS, road injury, diarrheal diseases, self-harm, iron-deficiency anaemia and skin diseases are among the top causes of morbidity and mortality.[5–7] Identifying adolescents with poor health, health-compromising behaviours or undiagnosed disability is important given (a) the growing

### STRENGTHS AND LIMITATIONS OF THIS STUDY

⇒ This study will utilise existing healthcare infrastructure in low-income and middle-income country settings, assessing real-world implementation situations and therefore it will be relatively straightforward to directly apply the findings to programmes.

⇒ This is a relatively large study of 6000 adolescents in three countries. The study takes the views of young people centrally into the design of the intervention.

⇒ Although the primary outcome is an implementation science/programmatic outcome, the effectiveness data is based on pre–post comparison.

⇒ This study will have limited ability to assess sustainability of effects over the longer term as the follow-up period is 4 months.

⇒ This study is operating in three African cities which may limit generalisability to rural areas.

number of adolescents and their low frequency of regular contacts with health services,[8] (b) the high proportion of the total global burden of disease that occurs in adolescence, (c) the fact that many key health conditions (eg, mental health disorders) and behaviours (eg, tobacco and alcohol use, unhealthy diet, low physical activity and risky sexual behaviours) that predispose to preventable serious conditions in later life start in adolescence, (d) the negative impact of poor health on educational attainment and employability and other transitions to healthy adulthood and (e) gender-related vulnerabilities, including violence, abuse, unintentional injury, sexual and reproductive health (SRH) and gendered mental health outcomes which may emerge or be exacerbated during this period of life, setting negative trajectories to lifetime and intergenerational health and well-being.[4]

Systematic reviews have identified individual interventions that are effective at improving various aspects of adolescent health and/or well-being.[4] However, most adolescents only come into contact with health services when they are ill, and services are not always appropriate for their needs.[9] This represents a missed opportunity for early detection of health problems, for health promotion and for the development of health-seeking behaviours. Early and sustained engagement with health and social services could reap a triple dividend for human development by improving the health and well-being of adolescents, their health and well-being in adulthood and the health and well-being of their future offspring.[2 4 10]

Routine health and well-being check-up visits for adolescents that screen for multiple conditions and risk behaviours could provide an entry point into services and be highly cost-effective.[11 12] Obtaining evidence on the optimum content, delivery, effectiveness and cost of check-ups is a high priority for adolescent health research so that governments can be informed by the evidence on how to initiate or strengthen existing health and well-being check-ups during adolescence.[13] Many high-income countries have national recommendations related to adolescent health check-ups, which have been largely based on expert opinion.[14 15] In LMICs, if provided at all, preventive and promotive health services for adolescents are largely provided in schools and are usually limited to deworming and vaccination campaigns. They do not usually address other key conditions and risk factors such as nutrition, mental health, SRH or disability.[16 17] If a system-wide approach to check-ups exists in adolescence, in LMICs, it is often limited to a screening activity without other components such as brief intervention or anticipatory guidance.[17]

This paper describes the protocol for the Y-Check: Evaluating the effects of adolescent health check-ups study, a prospective hybrid implementation-effectiveness study evaluating the feasibility, acceptability, short-term effects, costs and cost-effectiveness of the Y-Check intervention in three African cities. This study has received approval from the World Health Organisation (WHO/ERC Protocol ID Number ERC.0003778); Ghana Health Service (Protocol ID Number GHS-ERC: 027/07/22), the United Republic of Tanzania National Institute for Medical Research (Clearance No. NIMR/HQ/R.8a/Vol. IX/4199), the Medical Research Council of Zimbabwe (Approval Number MRCZ/A/2766) and the London School of Hygiene and Tropical Medicine (Approval Numbers 26 395 and 28312) .

### The Y-Check intervention

Y-Check is a novel intervention delivering a health and well-being check-up and, where indicated, will provide on-the-spot care and/or referral for common conditions on two occasions in adolescence (in young adolescents (10–14 year olds)—soon after the onset of puberty—and in older adolescents (15–19 year olds)—when many adolescents become, or are soon to become, sexually active). It will also provide health promotion information and materials to support positive behaviours and healthy lifestyles during adolescence and beyond. The intention is that in the context of a future routinely delivered programme, every adolescent will have two guaranteed contacts with the healthcare system. Adolescents will only be screened for conditions that have an accurate, low-cost, acceptable screening test and a locally accessible, effective intervention. The conditions selected for screening will be chosen to reflect the local epidemiological contexts (eg, screening for malaria will only take place in malaria endemic areas). Respecting specific requests from the Ministries of Education in all three cities, the study will only include SRH screening and services at the community sites (which only include older adolescents).

Figures 1 and 2 present the Theory of Change and description of the intervention. Table 1 applies the Template for Intervention Description and Replication (TIDieR) checklist[18] to describe details of the intervention.

Locally accessible services will be identified and assessed in terms of their ability to provide the services recommended by local and WHO guidelines, willingness to accept referred adolescents, and the fees charged to the project will be negotiated by the research team for services provided to referred adolescents (where adequate services are not covered by national health insurance schemes, free NGO services or free public healthcare).

## METHODS/DESIGN
### Aims

The aim of the study is to develop and implement in three African cities a potentially sustainable adolescent health check-up programme, and evaluate the acceptability, feasibility, short-term effects and cost-effectiveness of the programme to improve health and well-being. The study was launched in September 2021 and will run until June 2025.

**Figure 1** Theory of Change for Y-Check, an adolescent health and well-being check-up.

## Objectives

1. To develop and pilot test a check-up programme for adolescents that screens for important preventable and treatable health conditions using accurate and acceptable screening tests and provides locally accessible effective interventions.
2. Through a prospective intervention study in selected schools and communities to:
   – Estimate short-term impacts on adolescent health and well-being outcomes: clinical outcomes, health-related knowledge and behaviours, intentions,

agency and perceived social support for behaviour change; engagement with health services.
   – Understand, through process evaluation, the feasibility and fidelity of implementation, the acceptability and uptake and the influence of context.
   – Estimate the cost-effectiveness of the programme in reducing overall disease burden and improving adolescent well-being
3. Obtain information on key parameters needed for the planning of an evaluation study: prevalence of health conditions and behaviours, acceptability

**Figure 2** The Y-Check intervention package. *The intervention package may vary according to setting.

**Table 1** Template for Intervention Description and Replication (TIDieR) checklist describing the Y-Check intervention

| Item | Item |
|---|---|
| **Brief name** | |
| 1 | Evaluating the effectiveness of adolescent health check-ups (Y-Check) |
| **Why?** | |
| 2 | Identifying adolescents with poor health, health-compromising behaviours or undiagnosed disability is important for their health and well-being, and also for communities and nations |
| | Most adolescents only come into contact with health services when they are ill, and services are not always appropriate for their needs |
| | Routine health and well-being check-up visits for adolescents that screen for multiple preventable and/or treatable conditions and risk behaviours could provide an entry point into services and be highly cost-effective |
| **What?** | |
| 3 | The intervention includes a comprehensive health check-up for priority conditions customised to national and local contexts. |
| | Where indicated, Y-Check will provide on-the-spot care and cover all clinical costs associated with referrals to further care provided by the public health system or non-governmental organisations (NGOs). |
| | During the check-up, adolescents will receive health promotion information and limited supplies of key health commodities. |
| | Clinical costs of services are covered by the study if accessed within 4 months of the check-up. |
| 4 | Adolescent-friendly services will be provided, as defined by WHO (2018). Nationally approved protocols will be applied. Adolescent privacy and confidentiality will be protected. |
| **Who provided?** | |
| 5 | Y-Check teams will be staffed with health professionals trained to provide quality adolescent-friendly health services in line with nationally approved protocols. Y-Check teams will also be trained in the use of the digital application which will be used for data collection. Public and private not-for-profit care facilities providing referrals will meet national accreditation guidelines. |
| **How?** | |
| 6 | The Y-Check service will take place over a 60–90 min period face-to-face. Any referrals will only be subsidised by the study if they take place within 4 months. |
| **Where?** | |
| 7 | The Y-Check service will be provided in schools and community venues, in outdoor tents where required. |
| | Referrals will be to public or private not-for-profit providers as close as possible to the adolescent's home. Providers will be vetted by the study team as being able to provide the necessary referral services to national and WHO-recommended standards. |
| **When and how much?** | |
| 8 | Within the current phase of the study, each adolescent will receive Y-Check once. Within a routine programme, the intention would be that the intervention will be delivered twice during adolescence, once when the adolescent is 10–14 years old, and a second time when they are 15–19 years old. |
| **Tailoring** | |
| 9 | The content of the intervention is tailored to local context. The exact set of conditions that will be assessed as part of Y-Check will be adapted based on burden of disease, and availability of local tests and referral services. |
| **Modifications** | |
| 10 | Any modifications will be reported in the article reporting the results of the study. |
| **How well?** | |
| 11 | Intervention fidelity (adherence, integrity and quality) will be evaluated through a process evaluation including youth-friendly health services quality. |
| 12 | Intervention fidelity will be reported in the article reporting the results of the study. |

of referral, feasibility of following-up programme participants and delivering quality follow-up care, initial estimates of the impact of the programme on longer term health, educational and well-being outcomes based on the short-term implementation and effectiveness outcomes observed in this phase of the research programme, and factors related to the optimal implementation of the Y-Check intervention.

4. To refine the programme and its Theory of Change, and finalise optimal methods for the measurement of the impact of the programme in future studies.

 Banati P, *et al. BMJ Open* 2024;**14**:e077533. doi:10.1136/bmjopen-2023-077533

## Patient and public involvement

The intervention was designed following formative research conducted in three African countries between 2019 and 2020.[19–21] This formative research revealed that the proposed adolescent health and well-being check-ups are likely to be feasible to implement and acceptable to stakeholders in Ghana, Tanzania and Zimbabwe, and are likely to meet the perceived needs of key stakeholders including adolescents, their parents and key policy makers in the health and education sectors.[22] Further, we showed that the programme is likely to produce a substantial yield of important, previously untreated, treatable conditions. Human-centred design techniques were used alongside desk review to define elements of objective and subjective importance to the health and well-being of adolescents, identify facilitators and barriers to adolescent health seeking, preferences for delivery of routine health check-ups, and potential effects of interventions to select the content and method of delivery of the Y-Check intervention. Interviews and participatory workshops with adolescents, parents of adolescents and key stakeholders from the ministries of health and education, non-governmental organisations, healthcare workers and teachers found that there was overall support for the introduction of routine health check-ups.[19–21] To navigate potential barriers, stakeholders suggested clear messaging, awareness building and sensitisation campaigns to overcome disinterest in preventative healthcare and, in some contexts, mitigate cultural or religious messaging against healthcare engagement.[19]

## Theory of Change

We hypothesise that a routine health and well-being check-up visit for adolescents that screens for multiple conditions and risk behaviours will have an immediate and long-term positive impact on health and well-being outcomes (figure 1).

Health seeking and promotion behaviours among adolescents operate in complex environments and across ecological levels,[10] with determinants at individual, interpersonal institutional/organisational, community and public policy levels. Drawing from the health promotion literature,[23 24] the Theory of Change for Y-Check (figure 1) draws on thinking that recognises predisposing, enabling and reinforcing factors as capacities to be strengthened in order to achieve adolescent well-being at the individual level; that responsive parenting can support adolescents to meet their own health and well-being goals; that systems-based approaches (including stronger linkages between health and education systems) can improve outcomes for adolescents, especially reaching the most vulnerable and those in need and that an enabling environment (especially in schools and communities) can support adolescents to take action towards improving their health.

## Study setting

Our study will be undertaken in three African cities: Cape Coast in Ghana, Mwanza in Tanzania and Chitungwiza in Zimbabwe. These cities are described in table 2.

**Table 2** The study cities, schools and communities

| Cape Coast, Ghana | Mwanza, Tanzania | Chitungwiza, Zimbabwe |
| --- | --- | --- |
| Cape Coast Metropolis is located on the coast of Ghana, 150 kms west of the capital city, Accra. It has a population of 169 894 with three quarters of the households residing in urban areas. Literacy in 11–24 year olds is about 97%. In 2016, 11 233 (68.8%) of 12–14 year olds were enrolled in junior high schools while 8407 (91.6%) of 15–17 year olds were enrolled in senior high schools. For Ghana as a whole, primary and secondary net enrolment rates in 2019 were 86% and 57%, respectively.[38] There are 36 health facilities (26 public and 10 private) in the metropolitan area, including a regional hospital that serves as a secondary referral facility. The study will be conducted in eight schools and local community venues in four communities that include two relatively affluent communities with trading being the main source of livelihood and two relatively poorer communities where fishing and farming dominate, respectively. | Mwanza is located on the southern shores of Lake Victoria in North-Western Tanzania and is the second largest city in Tanzania with a population of over 900 000 and an annual growth rate of 3%.[39] Economic activities in Mwanza include fishing and fish processing, subsistence agriculture and support services to nearby gold and diamond mines. Adolescents make up 24.2% of the population of the city (Tanzania National Bureau of Statistics, 2016). As of 2020/2021, the primary and secondary school net enrolment rates were 82% and 39%, respectively.[39] Available public health services include 26 dispensaries, 5 health centres, 2 district hospitals, 1 regional hospital and 1 tertiary/teaching hospital.[39 40] The study will be conducted in 4–6 purposively selected communities and in up to eight primary schools and eight secondary schools within the catchment area of health facilities serving the selected communities in the two districts within Mwanza city. | Chitungwiza is the third largest city in Zimbabwe, located approximately 25 km south of the capital city, Harare. It has a population of about 456 000.[41] The houses are mostly high-density, single-story, detached units with small yards that are generally used for growing vegetables. Most of the people work in Harare, as there is little industry in Chitungwiza itself. Zimbabwe has a school-going population (8–18 years) of approximately 4.3 million.[42] Net primary enrolment rate across Zimbabwe is 94%; net secondary enrolment rate is 54%.[41] In Chitungwiza, there is one tertiary hospital, 4 public primary healthcare facilities, 20 private medical facilities, 30 government primary schools and 13 government secondary schools (all mixed sex). The study will be conducted in four distinct communities which are representative of the urban, peri-urban and rural populations of Chitungwiza. Eligible schools must have a student population of at least 200 learners in grade 6 or at least 75 learners in Form 5; and be located in or close to one of the selected study communities. |

## Study design

In this prospective hybrid implementation-effectiveness study, 2000 adolescents per city who receive the Y-Check intervention will be followed up at 4 months, and at 12 months (Zimbabwe only).

## Stakeholder engagement

In each city, the research study is undertaken in partnership with both the national and municipal Ministries of Health and Education. Each country has a policy framework that provides encouragement for the introduction of health and nutrition education and promotion among adolescents, including screening for communicable and non-communicable diseases, immunisation, growth monitoring and assessments and nutritional services.[25–27]

This study will build on stakeholder engagement, the process for which was established in each research setting during the formative phase. In each city, a Community Advisory Committee (CAC) comprising key community leaders and stakeholders will be reinforced or set up to facilitate input from, and feedback to, participating communities and a Youth Advisory Group (YAG) will provide a forum for adolescents to input into the programme. The YAG will meet with research staff at least four times per year, be active participants in programme design and dissemination workshops and help to ensure that the programme meets the needs of adolescents. Community engagement will be an ongoing process through regular contacts with the CAC, the YAG and other stakeholders, such as teachers, health workers, Community-Based Organisations (CBOs), Non-Governmental Organisations (NGOs) and religious leaders. In addition, a key aspect for building confidence within communities is the knowledge that the study has the support of the government.

## Intervention development and pilot testing

Prior to implementation, preparatory activities will include community engagement, participatory co-design, negotiating referral arrangements and pretesting of screening tools, procedures and referral protocols. Pilot studies in each setting will provide initial estimates of the frequency of health and behavioural outcomes, and help to refine the intervention model.

Pilot testing will involve the implementation of the screening tools and procedures with approximately 200 adolescents in each of the three cities with revisions and repeat pilot testing where required. Adolescents who participate in the pilot study will be excluded from the main study if the procedures change following the pilot. There will be an opportunity for young people and stakeholders to suggest additional client-centred outcomes that may reflect some of their priority concerns or intentions that should be captured.

## Intervention implementation

The intervention will be delivered over a period of 2–6 months in each of the settings. The follow-up visits will take place at the same school or community setting as the initial check-up. In addition to covering all clinical costs, the equivalent of US$5 will be given to each participant who attends the follow-up to cover any transport costs that they might have incurred. Additionally, health and hygiene related items will also be provided for adolescents to take home, including tooth cleaning kit (toothbrush and toothpaste), fruit, bottle of water, two pairs of underpants and pack of reusable sanitary pads (girls only).

## Composition and training of Y-Check team

The Y-Check team will be trained to deliver adolescent-responsive and age-appropriate services according to national and WHO guidelines, recognising also the needs for privacy and confidentiality.[28] This includes providing services that are attractive to adolescents, meet their needs comfortably and responsively and that are attentive to their privacy. These principles and approaches will be embedded into each part of the Y-Check intervention. Visual and auditory privacy will be prioritised, through the use of separate tents, rooms or screens. Health workers will employ standard gowning and draping for clinical procedures.

For infection prevention and control, all study procedures including interviews, physical examinations and blood tests will take place in well-aerated tents or outdoors, and will follow relevant nationally approved protocols for all staff and participants.

The Y-Check team will be trained in good clinical practice, data protection and confidentiality and clinical staff will be trained in counselling for participants testing positive for any of the conditions being screened for within Y-Check as well as in general counselling skills.

## Inclusion and exclusion criteria

To be included in the study, adolescents aged 10–19 years must fall into one of the first three categories below and fulfil category 4.

1. Be attending selected classes of year 5 of primary school in Mwanza (median age 11 years); grade 5/6 of primary school in Chitungwiza (median age 11 years) or year 1 of Junior Secondary School in Cape Coast (median age 12 years) OR
2. Be attending selected classes in year 3 of Secondary School in Mwanza (median age 17 years), Form 3/4 in Chitungwiza (median age 17 years) or year 2 of Senior Secondary School in Cape Coast (median age 16 years) OR
3. Be resident in a selected community during the time of the Y-Check intervention, and be aged 16–19 years AND
4. Have a completed and signed Informed Consent form, or a signed Informed Assent Form and signed Parental/Guardian Informed Consent Form if the adolescent is seen in the community and is below the national age of consent or is seen in a school, irrespective of their age.

## Consent and assent procedures

Before the visit of the implementation team, information on the Y-Check programme will be distributed to parents/guardians through the schools and to community members through an active communication campaign in collaboration with the CAC and the YAG. School and community meetings will allow parents and community members to ask questions about the programme and give their feedback.

In schools, adolescents will have a short introductory meeting with a member of the Y-Check team typically in a class or group setting. Parent meetings will then be held in each of the schools, to which all the parents and guardians of eligible learners will be invited. During these sessions, information will be provided about the study, its objectives and procedures, possible risks and procedures that will be used to maintain confidentiality. These meetings will provide an opportunity for the adolescents, parents and guardians of eligible adolescents to learn more about the Y-Check intervention and the research linked to it and to have their questions answered.

No participants will be screened, receive care or be counselled or interviewed without their informed consent (community participants who are above the national age of consent), or, for minors, their assent and parental consent, unless they are determined to be emancipated minors.[29] Following advice from Ministries of Education in all three countries, all adolescents seen in schools will be considered to be minors and require parental consent, irrespective of their age.

Minor adolescents' assent will be ascertained and documented in an assent form. Parents or guardians who would like their adolescent to receive the check-up will be asked to provide their written consent. On the day of the check-up visit, a verbal confirmation of their previous written assent will be requested from the adolescent. In Ghana and Tanzania, where the minimum age for providing consent to medical and health-related research is 18 years, clients of all ages under 18 will provide completed parental consent forms and provide written assent before proceeding through the check-up visit regardless of whether the check-up is in schools or communities. In Zimbabwe, a waiver of parental consent has been given by the Medical Research Council of Zimbabwe (MRC-Zimbabwe) so that participants aged 16 and 17 years who attend the check-ups in the community venues will be allowed to provide written consent for themselves.

The intervention will be conducted in private and not in the presence of the parent or guardian. Contact details of the study team will be shared with participants in case they have questions at a later stage. All participants will be reminded that participation is entirely voluntary and will be told that they can opt out of the research or services at any time.

## Data collection

### During the Y-Check intervention and follow-up

Data collection during baseline and follow-up visits will include self-completed evaluation questionnaires, self-reported screening tool responses and screening visit consultations, measurements and specimen collection and an exit interview. Data on the implementation process and on adolescent outcomes will be collected in digital and paper-based formats. A user-friendly digital data collection app for the check-ups will be developed and housed on a tablet computer for direct use by the adolescent. Initial sections will include audio-assisted, user-friendly self-completion questions for adolescents to fill out. This will utilise engaging content and processes, tailored to adolescents' interests. The option of a face-to-face interview will also be available if the adolescent is unable to use the tablet or has low literacy level. Health services registers and school registers will also be reviewed to determine the number of adolescents of the relevant age ranges, and school attendance by the classes involved in Y-Check. To help build the referral process, existing adolescent services will be mapped in the study communities.

### Process evaluation

The process evaluation is guided by the UK MRC's Process Evaluation framework to understand intervention implementation (including feasibility and fidelity), mechanisms of impact (including acceptability and uptake) and the influence of context.[30] Key implementation outcomes of interest are acceptability, adoption, appropriateness, feasibility and fidelity. Data on contextual factors and barriers and facilitators to programme implementation will be gathered using routinely collected programme monitoring data. Qualitative data will be collected through (1) observations of the Y-Check intervention and referrals, as well as team meetings; (2) in-depth interviews with eligible adolescents who received, adolescents who were referred and adolescents who did not receive Y-Check, as well as with school authorities and the Y-Check service providers and (3) participatory workshops with teachers, adolescents and parents. Quantitative programme monitoring data will be collected routinely within the Y-Check visit, including through a participant exit interview. Process evaluation data will be analysed iteratively and thematically, through regular analytical discussions and analytical memos to draw out the main themes emerging from the data. Across the pilot and intervention studies, data collection for the process evaluation will include real-time feedback to the implementation team.

### Economic evaluation

A costing study will be conducted to estimate the total costs of developing, setting up and running the Y-Check package, in school and community settings. A combination of top-down and ingredients-based costing approaches will be used to generate cost estimates for the whole package, and for each component/activity. All costs

will be estimated from the perspectives of the adolescents, the schools/community and implementing partners/service providers. Financial and economic costs will be calculated for all inputs. These inputs will be identified and measured using process data, staff interviews and observations, document review and accounting records.

Costs will be inputted and analysed in an Excel-based costing tool. The cost analysis will describe the distribution of costs across different forms of inputs, and will estimate the unit cost per adolescent reached, screened and treated on the spot or referred; cost per unit of measure for selected process and effect outcomes such as cost per condition detected, cost per condition appropriately treated on-the-spot or with a completed referral within 4 months, cost for a unit improvement in reported quality of life and Disability Adjusted Life Years averted.

The cost and cost-effectiveness estimates will be compared with other programmes in the region (eg, human papillomavirus vaccination and deworming) and will inform programme replication, scalability and financial sustainability.

### Data protections

Data protection will be strictly observed. After study completion, data will be stored in the LSHTM-curated digital repository 'Data Compass' following General Data Protection Regulation (GDPR) guidelines. Data and code registered in LSHTM Data Compass will be made open access following deposit. A Data Safety and Monitoring Board (DSMB) has been constituted to assist in managing adverse events, though we expect these to be very rare since all treatment and care are standard with no novel treatments.

### Study outcomes

Outcomes will be ascertained during the check-up screening visit and through collection of referral vouchers from the referral health facilities, and, for outcomes related to health and well-being impacts, through data from the 4 month and, in Zimbabwe only, 12 month follow-up visits. Outcomes related to completed referrals will be triangulated against participants' self-reports at the 4 month and in Zimbabwe only, 12 month follow-up visits. Review of school and health service registers will be used to see whether attendance has increased during the period when Y-Check is being implemented.

The primary outcome will be the proportion of those screening positive for at least one condition who receive appropriate on-the-spot care or complete appropriate referral for all identified conditions within 4 months. This will be measured using data collected at the initial check-up visit and through recovery of referral vouchers given to participants to allow them to access referral services for free during the 4 months after the Y-Check screening. Completed referral is defined as attending at least the first referral appointment.

Secondary implementation outcomes will include the proportion of those screening positive for each condition who receive appropriate on-the-spot care or complete appropriate referral for that condition within 4 months, the yield of previously untreated conditions, clinical outcomes at 4 months among those who had originally screened positive for each condition, and intervention acceptability, adoption, appropriateness, feasibility, fidelity and cost. Secondary effectiveness outcomes will include knowledge about health services and health behaviours, self-reported agency and self-efficacy to make decisions about their health, self-reported health-related risk and protective behaviours, reported engagement with health services, well-being, self-esteem and quality of life, clinical outcomes and educational outcomes, which will be collected within the Y-Check and follow-up visits. The short-term cost-effectiveness of the intervention will be estimated (calculated by a comparison of the costs of the intervention against the primary and secondary outcomes and including short-term changes in self-reported quality of life). All outcomes for the study are described in table 3.

### Sample size

In each city, the intervention will be implemented for 10–14 year olds in up to six government primary schools (n=500 for young adolescent girls, and n=500 for young adolescent boys), and for 15–19 year-olds in up to eight secondary schools and up to three community venues (n=500 for older adolescent girls, and n=500 for older adolescent boys), giving a total sample size of 2000 adolescents (10–19 y).

The sample size provides specified precision around the primary outcome. For example, for the primary outcome, within each age group and gender, if 150 (30%) of 500 participants screen positive for at least one condition, and 75% of those who screen positive are correctly managed (n=112), the 95% CI for correct management will be ±7%. The primary outcome used data from the initial check-up visit and referrals and did not require the 4 month follow-up data.

### Statistical analysis

All primary analyses will be conducted separately by study city; Cape Coast, Chitungwiza and Mwanza. Where comparable, secondary analyses will be conducted with the data from all three cities combined.

In our study sites, a contemporaneous comparison group is not required since no routine screening is currently taking place, and as a result, assessments at baseline will serve as the counterfactual for internal comparisons. Similarly, since there is no routine screening and treatment provided to adolescents of the target ages in the study population, a before–after comparison is appropriate since it is plausible to assume that reductions in the prevalence of the chronic conditions between the original Y-Check visit and the follow-up at 4 months will be due to the interventions provided through Y-Check.

We will follow Strengthening the Reporting of Observational Studies in Epidemiology (STROBE) guidelines for

**Table 3** Study outcomes and means of verification

| Outcome | Sources of data |
|---|---|
| **Primary outcome** | |
| Proportion of those screening positive for at least one condition who receive appropriate on-the-spot care or complete appropriate referral for all identified conditions within 4 months (ie, they attend a provider for referral care who has been accredited by the study team and has been shown to be capable of providing appropriate referral care). | ▶ Programme monitoring data including records of attendance for referrals.<br>▶ Screening tool (self-reported symptoms or conditions, measurements and clinical actions). |
| **Secondary outcomes** | |
| **Implementation outcomes** | |
| Proportion of those screening positive for each condition who receive appropriate on-the-spot care or complete appropriate referral for that condition within 4 months. | ▶ Programme monitoring data including records of attendance for referrals<br>▶ Screening tool (self-reported symptoms or conditions, measurements and clinical actions). |
| The yield of previously untreated conditions. | ▶ Programme monitoring data including records of attendance for referrals.<br>▶ Screening tool (self-reported symptoms or conditions, measurements and clinical actions). |
| Intervention acceptability (satisfaction): acceptability to adolescents and to other stakeholders (eg, schools, parents and health workers).<br><br>Intervention adoption (uptake, utilisation): Y-Check uptake, referrals completed.<br><br>Intervention appropriateness (perceived fit, perceived relevance and perceived usefulness): perceived value of the intervention to adolescents and to other stakeholders.<br><br>Intervention feasibility (actual fit, practicability): Y-Check visits completed, referrals completed, stakeholder support (including community). | ▶ Programme monitoring data including records of attendance for referrals.<br>▶ Screening tool (self-reported symptoms or conditions, measurements and clinical actions).<br>▶ Self-completed evaluation questionnaire.<br>▶ Exit interviews.<br>▶ Observations of the Y-Check visits and of selected referrals.<br>▶ Interviews and workshops with adolescents, healthcare providers, community members, teachers, parents and key stakeholders. |
| Intervention fidelity (adherence, integrity, quality): completeness of training for and delivery of intervention components; diagnostic accuracy; youth-friendly health services quality assessment. | ▶ Interviews and workshops with adolescents, healthcare providers, community members, teachers, parents and key stakeholders.<br>▶ Observations of the Y-Check visits and of selected referrals, including youth friendly services.<br>▶ Self-reported screening tool |
| **Economic outcomes** | |
| Cost of setting up and running the intervention.<br><br>Cost per adolescent with a newly diagnosed condition (overall and by condition).<br><br>Cost per adolescent with a newly diagnosed condition who received appropriate on-the-spot care or who completed an appropriate referral within 4 months (overall and by condition).<br><br>Short-term (4 months) cost-effectiveness: cost per improvement in health or well-being (eg, cost per case addressed or cured), cost per unit improvement in QALYs and per DALY averted. | ▶ Y-Check documentation and financial records<br>▶ Interviews with Y-Check staff and staff of the referral facilities.<br>▶ Programme monitoring data including records of attendance for referrals.<br>▶ Screening tool (self-reported symptoms or conditions, measurements and clinical actions). |
| **Client outcomes** | |
| Knowledge about health services and health behaviours.<br><br>Intentions to adopt healthy behaviours.<br><br>Agency to make decisions about health and well-being.<br><br>Perceived social support for behaviour change.<br><br>Health-related risk and protective behaviours.<br><br>Improvement in previously diagnosed health and well-being conditions.<br><br>Engagement with health and other services within the past 4 months.<br><br>Self-esteem.<br><br>Self-perceived well-being.<br><br>Quality of life.<br><br>Clinical outcomes. | ▶ Programme monitoring data including records of attendance for referrals.<br>▶ Screening tool (self-reported symptoms or conditions, measurements and clinical actions).<br>▶ Self-completed evaluation questionnaire. |
| Educational outcomes (eg, school attendance). | ▶ Self-completed evaluation questionnaire.<br>▶ School register review. |
| Client-defined outcomes (to be determined). | ▶ Self-completed evaluation questionnaire.<br>▶ Exit interviews. |

DALYs, Disability Adjusted Life Years; QALYs, Quality Adjusted Life Years.

the reporting of cohort studies. Descriptive analyses will be used to compare the community-level and school-level characteristics of the study communities and schools.

Quantitative programmatic data, including screening test results, services delivered and referrals made and completed will be reported by age, sex and city. The primary outcome is a single proportion which will be presented with a 95% CI for each of the four target groups: 10–14 year-old males, 10–14 year-old females, 15–19 year-old males 15–19 year-old females.

Secondary outcomes which are measured at a single-time point will be presented in a similar way to the primary outcome. For outcomes which are measured at two or more time points, a before–after analysis will be conducted comparing differences in measures between the time points. The unit of analysis will be the individual. For clinical outcomes which are measured at two or more time-points, the initial check-up visit (baseline) will give the prevalence of untreated conditions which will represent the counterfactual. The prevalence of conditions at the 4 month follow-up visit will be formally compared with this counterfactual to estimate the short-term effects of the intervention in improving these clinical outcomes. For analysis of outcomes measured at two time-points, we will use mixed-effects logistic regression (binary outcomes) or linear regression (continuous outcomes) adjusting for individual-level clustering as a random effect and school/community as a fixed effect. Health service and client determinants of correct management of conditions at 4 months will be analysed using multivariable regression.

### Ethics and dissemination

Ethics clearance has been received from WHO (WHO/ERC.0003778) and from all country national ethics bodies. Protocol modifications will be shared with the WHO Ethics Review Committee and relevant national ethics boards. Results will be published in at least three country-specific peer-reviewed journal publications and one multicountry publication. There will also be videos, briefs, webinars and meetings to disseminate results. All data will be placed into an open access repository after deidentification and anonymisation to ensure confidentiality and participant privacy.

### DISCUSSION

Over the last decade, adolescent well-being has become a global priority.[5] School health is also a growing area of policy interest.[31] WHO guidelines on school health services note that along with health promotion, health education, preventive interventions (such as immunisations and mass drug administration), clinical assessment and health services management, health screenings within school learners are one of the key pillars in the delivery of comprehensive school health services.[16] Screening programmes such as Y-Check provide a unique opportunity to detect easily treatable, high-burden health conditions, refer those requiring medical attention, treatment and care, as well as to advise and encourage adolescents to engage in healthy behaviours.

In a 2015 review, school health services were found to exist in at least 102 countries though their content varied considerably across 16 areas including vaccinations, SRH education, vision screening, nutrition screening and nutrition health education.[32] If all types of screening were combined, they were the second most commonly reported intervention in school health services, second only to immunisation. A later systematic review found evidence of routine health check-ups of school age children having been reported in 86 countries worldwide.[17] Despite their widespread existence, little quality evidence exists on how to promote good health for adolescents in educational settings,[32] and even less for multicomponent school health services,[33] especially in LMICs.[34]

Good practices in conducting adolescent health or well-being screenings are rarely reported. In 2024, WHO released new guidance on well-child and well-adolescent visits, which will recommend expanding routine screening tests to also integrate other well-being dimensions through a broader evaluation of social risks, emotional state and individual and family resources delivered with context-specific recommendations at key moments during the first two decades of life.[35] The successful implementation of such guidance requires robust measurement of the effectiveness of preventive interventions in adolescence.[36]

Evaluation of the Y-Check intervention will incorporate implementation science and effectiveness research. Such hybrid designs have important advantages over conducting separate studies. These include the potential for quicker translation of intervention research findings into programmes, the development and selection of more effective implementation strategies and more useful information for decision makers.[37]

The process evaluation findings will provide guidance for the next stage of the programme and for potential future sustainable and scalable implementation by local health authorities should it prove successful. Data on the short-term changes in clinical and behavioural outcomes will be used as inputs to model both short-term and long-term health and social impacts and as inputs to sample size and power calculations for a third phase of the Y-Check research programme, which plans to undertake a rigorous population level evaluation of the impact of routine check-ups on adolescent health and well-being.

Through WHO's advice to member states, findings from the Y-Check study have the potential to shape the delivery of adolescent health check-ups globally including identifying the optimal number, content and delivery for these services. Y-Check will advance the field by providing some of the first rigorous information on the effects of a health screening programme in three African cities, assessing implementation, effectiveness, cost and cost-effectiveness outcomes.

**Author affiliations**
[1]World Health Organization, Geneva, Switzerland
[2]Stellenbosch University, Stellenbosch, South Africa
[3]University of Ghana, Legon, Ghana
[4]National Institute for Medical Research Mwanza Research Centre, Mwanza, United Republic of Tanzania
[5]London School of Hygiene and Tropical Medicine, London, UK

⁶Biomedical Research and Training Institute, Harare, Zimbabwe
⁷Department of Infectious Disease Epidemiology, London School of Hygiene and Tropical Medicine, London, UK

**Acknowledgements** The authors thank our Programme Advisors: Professor Simon Gregson, Dr Sachin Shinde, Professor Fred Binka, Dr Suzanne Petroni, Professor Audrey Pettifor, Prof Shane Norris, Professor James Hargreaves, Patience Manhibi and Aveneni Mangome. DSMB members are Professor David Mabey, Dr Andrew Abassa, Prof Fred Binka and Dr Nothando Ngwenya. The authors thank adolescent participants in three countries. The support of participating Ministry officials in Ghana, Tanzania and Zimbabwe is gratefully acknowledged.

**Collaborators** The Y-Check Research Programme Team in alphabetical order includes: Philip B Adongo, MA, PhD, Department of Social and Behavioural Sciences, School of Public Health, University of Ghana; Kenneth S Adde, MPhil, Department of Population and Health, University of Cape Coast; Evans Agbeno, FGCS, MPH, MBCHB, Department of Obstetrics & Gynaecology, University of Cape Coast; Patricia Akweongo, MA, PhD, Department of Health Policy Planning and Management, School of Public Health, University of Ghana; Tsitsi Bandason MSc, The Health Research Unit Zimbabwe/Biomedical Research and Training Institute; Sarah Bernays, PhD MA, Global Health and Development, London School of Hygiene and Tropical Medicine and School of Public Health, University of Sydney; Rudo Chingono PhD, The Health Research Unit Zimbabwe/Biomedical Research and Training Institute; Chido Dziva Chikwari PhD, MSc, BSc, The Health Research Unit Zimbabwe/ Biomedical Research and Training Institute and MRC International Statistics & Epidemiology Group, Department of Infectious Disease Epidemiology, London School of Hygiene & Tropical Medicine, London, UK; Ethel Dauya MPH, The Health Research Unit Zimbabwe/Biomedical Research and Training Institute; Samuel Derry, PhD, Department of Bio-Statistics, School of Public Health, University of Ghana; Eric Koka, MPhil, PhD, Department of Sociology and Anthropology, University of Cape Coast; Constance Mackworth-Young, PhD MSc, Global Health and Development, London School of Hygiene and Tropical Medicine; Salome Manyau MSc, The Health Research Unit Zimbabwe/Biomedical Research and Training Institute; Gerry Mshana PhD, Mwanza Intervention Trials Unit (MITU)/National Institute for Medical Research (NIMR); Bernard A Owusu, MPhil, Department of Population and Health, University of Cape Coast; Yovitha Sedekia, MPHDC, PhD, Mwanza Intervention Trials Unit (MITU)/National Institute for Medical Research (NIMR); Victoria Simms PhD, MRC International Statistics & Epidemiology Group, London School of Hygiene and Tropical Medicine, Hannah Taylor-Abdulai, MPH, MPhil, PhD, Department of Physician Assistant Studies, University of Cape Coast; Mandi Tembo, PhD, The Health Research Unit Zimbabwe/Biomedical Research and Training Institute and Department of Global Health and Development, Faculty of Public Health and Policy, London School of Hygiene and Tropical Medicine, London, UK.

**Contributors** PB, AMD and DAR conceived and drafted the paper. All authors contributed to writing. BW, SK, AMD, FN, FG and MKN provided country-specific inputs. GG contributed to the economic evaluation section. HAW contributed to the study outcomes and statistical analysis sections. VB and RAF contributed to the sections on consent, introduction and discussion.

**Funding** This work was supported by the Botnar Foundation grant number (RG21-001) and UKRI (MR/T043156/1).

**Competing interests** PB and VB are WHO staff members. The opinions expressed are theirs and do not necessarily represent the policies and positions of the WHO. The other authors declared no competing interests.

**Patient and public involvement** Patients and/or the public were involved in the design, or conduct, or reporting, or dissemination plans of this research. Refer to the Methods section for further details.

**Patient consent for publication** Not applicable.

**Provenance and peer review** Not commissioned; externally peer reviewed.

**ORCID iDs**
Prerna Banati http://orcid.org/0009-0003-2997-9224
Helen Anne Weiss http://orcid.org/0000-0003-3547-7936

Valentina Baltag http://orcid.org/0000-0002-6766-0842
Farirai Nzvere http://orcid.org/0000-0002-3260-6889
Mussa Kelvin Nsanya http://orcid.org/0000-0001-7653-4541
Aoife Margaret Doyle http://orcid.org/0000-0002-3305-7738

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
