## [Reviewer comments · BMJ Open]

ARTICLE DETAILS

TITLE (PROVISIONAL)	An adolescent health and wellbeing check-up programme in three African cities (Y-Check): protocol for a multimethod, prospective, hybrid implementation-effectiveness study
AUTHORS	Banati, Prerna; Ross, David Anthony; Weobong, Benedict; Kapiga, Saidi; Weiss, Helen; Baltag, Valentina; Nzvere, Farirai; Glozah, Franklin; Nsanya, Mussa; Greco, Giulia; Ferrand, Rashida; Doyle, Aoife

VERSION 1 – REVIEW

REVIEWER	Sargsyan, Zhanna American University of Armenia
REVIEW RETURNED	05-Dec-2023

GENERAL COMMENTS	I would like to congratulate the authors for an informative and well-written manuscript on the multi-country prospective intervention study assessing the implementation, effects and cost-effectiveness of Y-Check program using mixed methods of evaluation. Here are some minor comments for authors to consider: 1. You mentioned that by the request of Ministries of Education, sexual and reproductive health screening and services will be provided to the older adolescents only. What is the rationale behind this request? Please elaborate why you will not screen the younger adolescents and provide materials for those topics. (Lines 37-41)2. Lines 27- You mention that the YAG will meet the research team 4 times during a year. However, it is not clear the timeframe for the intervention development and pilot testing (Lines 44). Could you please specify the approximate timeline?3. Also, you mention that the pretest will be conducted in each city among 200 adolescents (pages 15-34, Lines 3-5). It was not clear to me if you will exclude those adolescents from the intervention or not. Please add the information.4. When calculating the sample size, did you also account for the lost to follow up adolescents? I wanted to extend my sincerest appreciation for the outstanding work you've presented in your manuscript. Your dedication to thorough research and clear articulation of ideas have made this piece an insightful and engaging read. Hope to see the results of the Y-CHEK and recommendations to scale-up for other LMICs.
---

REVIEWER	Tibber, Marc University College London, Department of Clinical, Educational and Health Psychology
REVIEW RETURNED	07-Feb-2024

GENERAL COMMENTS	This protocol describes a large-scale study in a very important area of research. I have no feedback to give other than that the paper is well-written and comprehensive, and describes the methodology, analyses, and outcomes clearly. I look forward to hearing about the findings of this study.
--

VERSION 1 – AUTHOR RESPONSE

Reviewer: 1

Ms. Zhanna Sargsyan, American University of Armenia

Comments to the Author:

I would like to congratulate the authors for an informative and well-written manuscript on the multi-country prospective intervention study assessing the implementation, effects and cost-effectiveness of Y-Check program using mixed methods of evaluation. Here are some minor comments for authors to consider:

Author comment: Thank you.

1. You mentioned that by the request of Ministries of Education, sexual and reproductive health screening and services will be provided to the older adolescents only. What is the rationale behind this request? Please elaborate why you will not screen the younger adolescents and provide materials for those topics. (Lines 37-41)

Author comment: Unfortunately, the Ministries did not provide a specific reason for their request that SRH screening and services be excluded from the in-school the Y-Check programme and included at community settings only. Requests to exclude these services in schools may have been due to fears about potential opposition from teachers and/or parents but we don't know for sure so we cannot add any additional clarification in the manuscript.

2. Lines 27- You mention that the YAG will meet the research team 4 times during a year. However, it is not clear the timeframe for the intervention development and pilot testing (Lines 44). Could you please specify the approximate timeline?

Author comment: This has been added. Please see page 10 lines 218-219

3. Also, you mention that the pretest will be conducted in each city among 200 adolescents (pages 15-34, Lines 3-5). It was not clear to me if you will exclude those adolescents from the intervention or not. Please add the information.

Author comment: This has been added on page 14 lines 288-290

4. When calculating the sample size, did you also account for the lost to follow up adolescents?

Author comment: The primary outcome requires data from the initial check-up visit and referral visits. Follow-up data collected at 4 months is not required for the primary outcome. We have added some text in the manuscript to clarify this on page 21 line 466.

I wanted to extend my sincerest appreciation for the outstanding work you've presented in your manuscript. Your dedication to thorough research and clear articulation of ideas have made this piece an insightful and engaging read. Hope to see the results of the Y-CHEK and recommendations to scale-up for other LMICs. Thank you.

Reviewer: 2
Dr. Marc Tibber, University College London

Comments to the Author:
This protocol describes a large-scale study in a very important area of research. I have no feedback to give other than that the paper is well-written and comprehensive, and describes the methodology, analyses, and outcomes clearly. I look forward to hearing about the findings of this study. Thank you.

Reviewer: 1
Competing interests of Reviewer: No Competing Interests to Declare.

Reviewer: 2
Competing interests of Reviewer: None

VERSION 2 – REVIEW

REVIEWER	Sargsyan, Zhanna American University of Armenia
REVIEW RETURNED	17-Apr-2024
GENERAL COMMENTS	Congratulations!